# Predictions from standard epidemiological models of consequences of segregating and isolating vulnerable people into care facilities

**Joseph Hickey**\*, **Denis G. Rancourt**

Correlation Research in the Public Interest, Ottawa, Ontario, Canada

\* joseph.hickey@alumni.ucalgary.ca

**Data Availability Statement:** All relevant data are within the manuscript and its Supporting Information files. This is a mathematical modeling study, and all details needed to reproduce the

## Abstract

### Objectives

Since the declaration of the COVID-19 pandemic, many governments have imposed policies to reduce contacts between people who are presumed to be particularly vulnerable to dying from respiratory illnesses and the rest of the population. These policies typically address vulnerable individuals concentrated in centralized care facilities and entail limiting social contacts with visitors, staff members, and other care home residents. We use a standard epidemiological model to investigate the impact of such circumstances on the predicted infectious disease attack rates, for interacting robust and vulnerable populations.

### Methods

We implement a general susceptible-infectious-recovered (SIR) compartmental model with two populations: robust and vulnerable. The key model parameters are the per-individual frequencies of within-group (robust-robust and vulnerable-vulnerable) and between-group (robust-vulnerable and vulnerable-robust) infectious-susceptible contacts and the recovery times of individuals in the two groups, which can be significantly longer for vulnerable people.

### Results

Across a large range of possible model parameters including degrees of segregation versus intermingling of vulnerable and robust individuals, we find that concentrating the most vulnerable into centralized care facilities virtually always increases the infectious disease attack rate in the vulnerable group, without significant benefit to the robust group.

### Conclusions

Isolated care homes of vulnerable residents are predicted to be the worst possible mixing circumstances for reducing harm in epidemic or pandemic conditions.

results are contained in the manuscript and supporting information.

**Funding:** The authors received no specific funding for this work.

**Competing interests:** The authors have declared that no competing interests exist.

## Introduction

During the COVID era (from the World Health Organization (WHO) 11 March 2020 COVID-19 pandemic declaration to present), many governments have imposed policies isolating or segregating people deemed highly vulnerable to respiratory disease, including by restricting movement into and out of long-term care homes where elderly and physically or mentally disabled people reside and reducing contacts between care home residents and staff [1–4].

Although it was known that isolation and loneliness can have serious negative health consequences for segregated vulnerable people [5–8], and although it was known that residents concentrated in care homes are particularly vulnerable to infectious diseases [8–12], and although data from the spring of 2020 showed disproportionately large all-cause mortality increases in long-term care homes that were positively correlated with the number of care home residents [13, 14], governments continued to implement policies confining vulnerable people into care homes and reducing social contacts with visitors and staff more than one year after the WHO's 11 March 2020 COVID-19 pandemic declaration.

Non-pharmaceutical interventions such as travel restrictions, workplace closures, and age-specific enforced social distancing or quarantining have been justified during the COVID era using theoretical infectious disease models based on the paradigm of spread by close-proximity pairwise contacts [15–19]. At their core, the baseline epidemiological models on which essentially all more sophisticated models are built, have two main parameters determining whether an infectious disease epidemic emerges and, if it does, its magnitude and duration. These two parameters are: the rate at which individuals experience pairwise contacts with others that could result in transmission of the infection, and the rate at which infected individuals recover and become immune. When modeling the consequences of any non-pharmaceutical intervention, it is crucial to first understand the impact of varying contact frequencies and recovery rates, before adding more sophisticated model features.

Researchers who have explored the impact of non-pharmaceutical interventions in models of COVID-19 spread with different age groups typically distinguish the age groups based on susceptibility to infection, contact rates, and the probability of severe outcome or death upon infection, but do not consider the impact of age-group-specific recovery rates [20–27]. When such models do allow for longer recovery times for older or more vulnerable individuals, including in models that specifically seek to represent vulnerable individuals residing in care homes, the models are highly-detailed in structure, involving many disease-state compartments and associated parameters [28–38]; the authors of these studies have not made comprehensive explorations of the model results across broad ranges of parameter values, but rather investigate model outcomes for different intervention scenarios using narrow ranges of epidemiological parameter values taken to be relevant to COVID-19.

However, in order to appreciate the spectrum of outcomes that are possible in a given theoretical model, and its limitations and sensitivity to assumptions, it is crucial to base the model on the simplest-possible sufficiently realistic conceptual foundation and only add extensions incrementally [39, 40]. This approach optimizes relevance and minimizes confounding the results with complexity and intangible propagation of error. Focusing on only the core model ingredients limits the dimensionality of the model, permitting the needed examination of the model's outcomes across a comprehensive range of parameter values. We adopt this exploratory and insight-generating approach, rather than an approach in which a more complex model is used to make predictions about the application of a specific policy. Detailed models geared toward specific policies should consider the insights and limitations identified in the baseline models.

Large-range exploration of the parameters is needed because the actual parameter values are not well delimited by empirical measurements and are often essentially unknown; and because unexpected effects or magnitudes of effects can occur in different otherwise unexplored and relevant regions of the parameter space.

We construct a simple susceptible-infectious-recovered (SIR) epidemic model consisting of two interacting populations, one representing the relatively robust majority of society and the other the vulnerable minority. The different health states of individuals in the two populations are represented by their different recovery times upon infection, as is well established for respiratory diseases [41, 42]. We investigate the size and duration of epidemics occurring for a broad range of different within- and between-population contact frequencies representing different segregation or isolation policy-linked behaviours. This approach allows us to make broad-ranging conclusions about the consequences of segregation of vulnerable people that apply to all epidemic models based on the SIR foundational assumptions.

## Model

We implement a susceptible-infectious-recovered (SIR) model for two populations, indexed as population "$r$" and population "$v$". The total number of $r$ individuals is $N_r$ and the total number of $v$ individuals is $N_v$, and the total population is $N = N_r + N_v$.

We assign the $r$ population to be the majority population of robust individuals, and the $v$ population to be the minority population of vulnerable individuals.

Following the usual SIR model structure, a person can be in one of three states: susceptible to infection (S), infectious (I), or recovered and immune (R). If a susceptible person comes into contact with an infectious person, the susceptible person can become infectious, and infectious people eventually recover. The numbers of susceptible, infectious, and recovered people in group $i$ (where $i$ can be either $r$ or $v$) at time $t$ are therefore $S_i(t)$, $I_i(t)$, and $R_i(t)$, respectively, and $N_i = S_i(t) + I_i(t) + R_i(t)$.

The number of individuals in each of the three epidemiological compartments, in each of group $r$ and $v$, evolve according to the following equations:

$$\frac{dS_r}{dt} = -S_r \left[ c_{rv}\beta_{rv}\frac{I_v}{N_v} + c_{rr}\beta_{rr}\frac{I_r}{N_r} \right] \tag{1A}$$

$$\frac{dI_r}{dt} = S_r \left[ c_{rv}\beta_{rv}\frac{I_v}{N_v} + c_{rr}\beta_{rr}\frac{I_r}{N_r} \right] - \gamma_r I_r \tag{1B}$$

$$\frac{dR_r}{dt} = \gamma_r I_r \tag{1C}$$

$$\frac{dS_v}{dt} = -S_v \left[ c_{vr}\beta_{vr}\frac{I_r}{N_r} + c_{vv}\beta_{vv}\frac{I_v}{N_v} \right] \tag{1D}$$

$$\frac{dI_v}{dt} = S_v \left[ c_{vr}\beta_{vr}\frac{I_r}{N_r} + c_{vv}\beta_{vv}\frac{I_v}{N_v} \right] - \gamma_v I_v \tag{1E}$$

$$\frac{dR_v}{dt} = \gamma_v I_v \tag{1F}$$

Eqs 1A–1F involve three sets of parameters, described below.

The parameters $\gamma_r$ and $\gamma_v$ represent the rates at which $r$ and $v$ individuals (robust and vulnerable individuals, respectively) recover from infection. Since the $v$ population represents the minority, vulnerable population: $N_v \leq N_r$. Since they are more vulnerable than $r$ individuals, $v$ individuals take a longer time to recover from infection, such that $\gamma_v \leq \gamma_r$.

We use a value of $\gamma_r = 75$ yrs$^{-1}$ corresponding to a recovery time of approximately 5 days for healthy individuals [43, 44], and we consider three values of $\gamma_v$, equal to $\gamma_r$, $\gamma_r/2$, and $\gamma_r/4$, corresponding to recovery times of approximately 5, 10, and 20 days for the $v$ individuals [41, 42].

The other two sets of parameters, $c_{ij}$ and $\beta_{ij}$, are intrinsically dependent, such that one set is actually redundant, which can be understood as follows. $\beta_{ij}$ represents the probability that a contact between a susceptible $i$ ($r$ or $v$) person and an infectious $j$ person results in infection of the susceptible $i$ person, whereas $c_{ij}$ represents the frequency (number per unit time) of contacts between an $i$ person and a $j$ person. Therefore, we are free to make the following simplification. Without loss of generality, in this paper we set $\beta_{rr} = \beta_{vv} = \beta_{rv} = \beta_{vr} = 1$. This means that the only contacts considered and counted are by definition contacts that are guaranteed to result in transmission when the contact involves a susceptible $i$ person and an infectious $j$ person.

There is no reason or advantage to considering other definitions of $c_{ij}$ having associated smaller values of $\beta_{ij}$; and it would make no difference in the calculated results arising from Eqs 1A–1F. Under this notational and conceptual simplification, the $c_{ij}$ are the dominant control parameters in the model, along with the recovery rates $\gamma_r$ and $\gamma_v$. We apply this interpretation of $c_{ij}$ (arising from setting all the $\beta$ parameters equal to 1) throughout the remainder of the paper.

The within-group contact frequencies, $c_{rr}$ and $c_{vv}$ are independent of one another. The between-group contact frequencies $c_{rv}$ and $c_{vr}$ are also independent. However, we impose the following relationship between $c_{rv}$ and $c_{vr}$, modulated by the coefficient $\lambda$:

$$c_{rv} = \lambda \frac{c_{vr} N_v}{N_r} \tag{2}$$

A value of $\lambda = 1$ corresponds to a strict proportionality between $c_{rv}$ and $c_{vr}$ determined purely by the relative sizes of the populations of the two groups, as would be common to impose in the sliding definition of contact in which $\beta_{ij}$ are undetermined [39].

In the present paper, $\lambda = 1$ effectively means that pairwise contact events that are of a physical proximity and duration sufficient to guarantee infection of a susceptible $v$ person by an infectious $r$ person are also sufficient to guarantee infection of a susceptible $r$ person by an infectious $v$ person. However, in principle, $\lambda$ can take values less than 1, due to the more resistant health status of $r$ individuals compared to $v$ individuals. Since, given the relative sizes of the populations $N_r$ and $N_v$, $c_{rv}$ is much smaller than $c_{vr}$ and typically much smaller than $c_{rr}$ in our analyses, we use a value of $\lambda = 1$ in the main text of this paper. In the S1 Appendix, we show that our results are robust against smaller values of $\lambda$.

We also define $c_r = c_{rr} + c_{rv}$ and $c_v = c_{vv} + c_{vr}$ to be the total contact frequencies of $r$ and $v$ people, respectively. The majority, robust ($r$) population is typically younger and more socially active than the minority, vulnerable ($v$) population, such that the frequency of all person-to-person contacts is generally higher in the $r$ group than the $v$ group [45]. However, when $c_r$ and $c_v$ represent the frequency of only those types of contacts that are guaranteed to result in infection of a susceptible individual (as per our simplifying assumption that $\beta_{rr} = \beta_{vv} = \beta_{rv} = \beta_{vr} = 1$, in the present article), then it is not unreasonable to consider that $c_v$ can be greater or significantly greater than $c_r$, due to the frailer health status of the $v$ individuals.

## Results

We examine the epidemic outcomes for the robust ($r$) and vulnerable ($v$) populations for a large range of possible contact frequencies and recovery rates. For specificity, we use a total population of $N = 10^7$ individuals, with a representative value $N_r/N = P_r = 0.95$, such that the $r$ population constitutes 95% of the entire society, and the $v$ population is 5%, approximately corresponding to the percentage of people over 80 years of age in Canada and in European Union countries [46, 47]. The simulations are "seeded" with 100 infectious individuals inserted proportionally into each of the two groups, such that $I_r(t = 0) = 95$ and $I_v(t = 0) = 5$.

The actual proportion of people residing in care homes or facilities in Canada is approximately 1.3% [48], which corresponds to a value of $P_r$ closer to 0.99. We verified that the results are the same on varying $P_r$ (including $P_r = 0.99$), $\lambda$, and seeding magnitude and distribution, which is shown in the S1 Appendix.

We define the attack rate among population $i$ as the proportion of initially-susceptible $i$ people who become infected during the epidemic:

$$A_i = (S_i(t_0) - S_i(t_f))/S_i(t_0), \tag{3}$$

where $S_i(t_0)$ is the number of susceptible $i$ people at the beginning of the epidemic and $S_i(t_f)$ is the number of susceptible $i$ people remaining once there are no longer any infectious people in either of the two groups ($r$ or $v$).

In order to examine the impact of policies that isolate or segregate the $v$ individuals from the $r$ group, we introduce the index $x$ equal to the share of a $v$ individual's contacts that are with $r$ people:

$$x = c_{vr}/c_v, \tag{4}$$

When $x = 0$, $v$ individuals only ever have contacts with other $v$ individuals, and when $x = 1$, $v$ individuals only ever have contacts with $r$ people. In this way, $x$, represents the degree of segregation versus intermingling of the $r$ and $v$ groups. Complete segregation is $x = 0$. Complete $r$-$v$ intermingling, while avoiding all $v$-$v$ contacts, is $x = 1$.

Fig 1 shows the evolution of the epidemic (number of new cases per day, over time) in the $r$ and $v$ groups, for different values of $x$. In this example, $c_r$ is slightly larger than $\gamma_r$ (in order that $c_r / \gamma_r$ ("$R_0$") $\approx 1.1 > 1$ such that an epidemic would occur in the $r$ group if it were completely isolated from the $v$ group) and $c_v$ is 25% larger than $c_r$. $\gamma_v = \gamma_r/4$, such that $v$ people take four times as long to recover from infection as $r$ people.

As can be seen in Fig 1, $x$ (the degree of separation or intermingling) has a large effect on the size and duration of the epidemics occurring in both the $r$ (robust, majority) and $v$ (vulnerable, minority) groups.

When $x = 0$, $v$ individuals only ever come into contact with other $v$'s, and the number of new cases per day in the $v$ group rapidly surges, peaks, and decays, and essentially all of the $v$ population becomes infected ($A_v \approx 1$, inset of Fig 1). An epidemic also occurs in the $r$ group, but the attack rate is smaller ($A_r$, inset) and it takes significantly longer for the epidemic to transpire (see the dashed blue line in the extreme lower-right corner of Fig 1).

In Fig 1, as $x$ is increased above 0, a larger and larger share of $v$ contacts are with $r$ individuals. In the $v$ group, the epidemic size (peak value of new cases per day and attack-rate) decreases with increasing $x$ and the duration of the epidemic increases. Going from $x = 0.5$ to $x = 0.75$ and $x = 1$, $A_v$ is significantly decreased, to the point where less than half of the susceptible, vulnerable $v$ population becomes infected. On the other hand, increasing $x$ above 0 initially increases $A_r$ and significantly shortens the time it takes for the number of new $r$ cases per

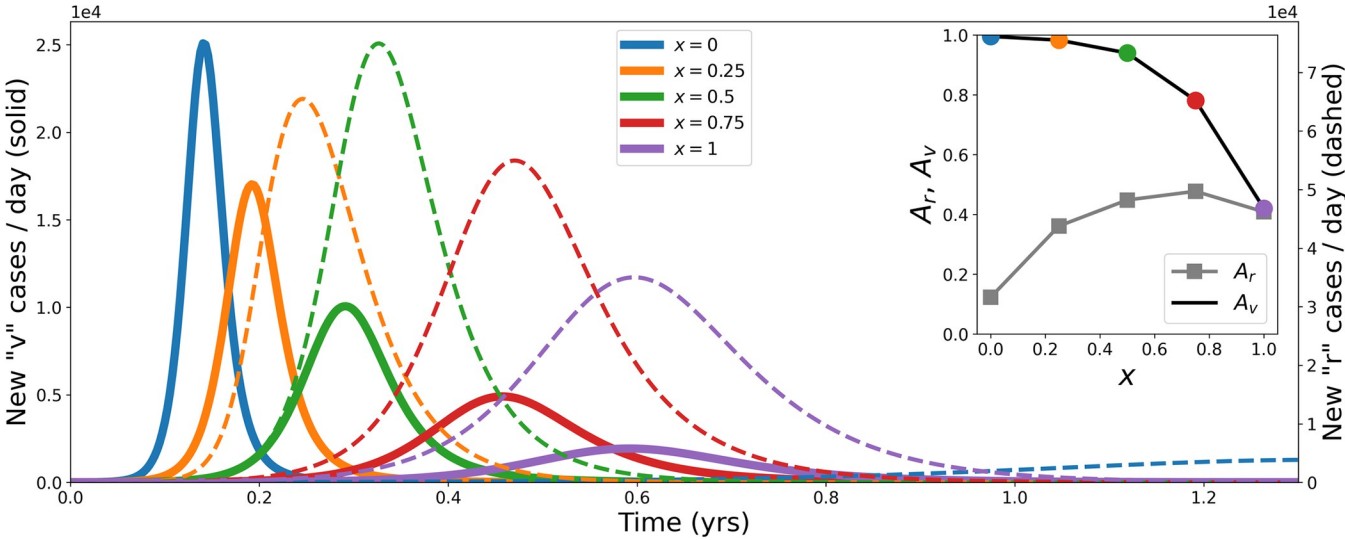

**Fig 1. Example epidemic curves for different values of the degree of segregation, $x$.** Epidemic curves showing the number of new cases per day in population $v$ (vulnerable, minority group, solid lines, left y-axis) and population $r$ (robust, majority group, dashed lines, right y-axis), for different values of $x$, and for the fixed model parameters indicated above the figure. Inset: attack-rates $A_r$ and $A_v$ as functions of $x$ (coloured circles indicate the $x$ values listed in the main figure legend).

day to surge and decay. When $x = 1$, the epidemic curves for the $r$ and $v$ populations have their peaks at approximately the same time, and the attack rates become similar for the two groups.

Fig 1 illustrates the important effect of $x$ on the epidemic outcomes in the two populations. In particular, it is apparent that larger $x$ (more contacts with robust individuals) can produce significantly better (lower attack rate) results for the minority vulnerable population. This is important if it is a general feature because the vulnerable individuals in the real world have higher risk of dying on being infected [49–51], which is the motivation for wanting to protect them. Larger values of $x$ result in epidemics of longer duration in the vulnerable population, but with significantly lower peak numbers of new cases per day and smaller attack rates. The attack rate in the robust population, $A_r$, increases with increasing $x$ up to a maximum around $x = 0.75$, then decreases as $x$ is increased to 1 (complete intermingling of $v$ with $r$). There is thus a trade-off that can occur, in which increasing $x$ decreases $A_v$ but increases $A_r$.

Next, we present figures showing results across our large range of possible and reasonable $c_r$ and $c_v$ values, for different degrees of segregation vs. intermingling, $x$, between the $r$ and $v$ groups, and for the different values of $\gamma_v$ representing different degrees of vulnerability of the $v$ population.

Fig 2 contains a collection of panels showing how the attack-rates $A_r$ and $A_v$ change as $c_r$ and $c_v$ are varied. Each panel corresponds to a choice of $x$ and $\gamma_v$.

The panel in the upper-left corner of Fig 2 corresponds to $x = 0$ and $\gamma_v = \gamma_r/4 = 18.75$. Since $x = 0$, there is complete segregation between the $r$ and $v$ groups. In this case, an epidemic emerges in the $r$ group when $c_r > \gamma_r$ and in the $v$ group when $c_v > \gamma_v$, and this can be seen by the fact that $A_r > 0$ when $c_r > 75$, for all values of $c_v$, and $A_v > 0$ when $c_v > 18.75$, for all values of $c_r$. Thus, when $x = 0$, we see the usual transition to an epidemic, which occurs in a one-population SIR model when $R_0 = c/\gamma > 1$, in each group.

The panels in the second through fifth rows of Fig 2 correspond to $x > 0$, progressively increasing up to $x = 1$ (fifth row). For many values of $c_r$, increasing $x$ results in a shift upwards (to higher $c_v$ values) of the red contour lines, indicating a decrease in $A_v$ for fixed $c_v$.

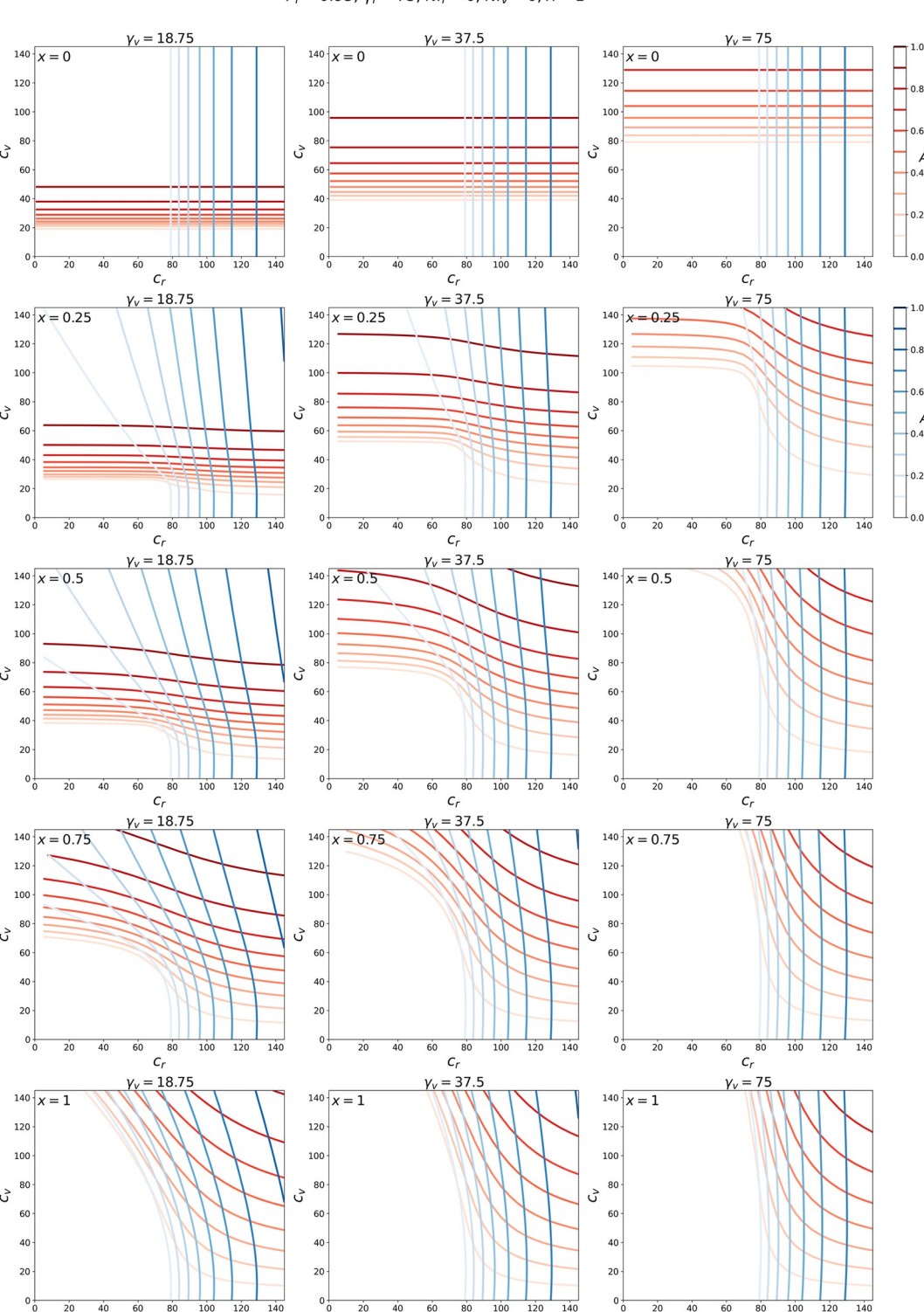

**Fig 2. Attack rate contour maps.** Contour maps of $A_r$ (blue lines, see scale at the upper right) and $A_v$ (red lines, see scale at the upper right) for a range of contact frequencies $c_r$ and $c_v$. Each column of panels corresponds to a different $\gamma_v$ and each row to a different $x$, as indicated.

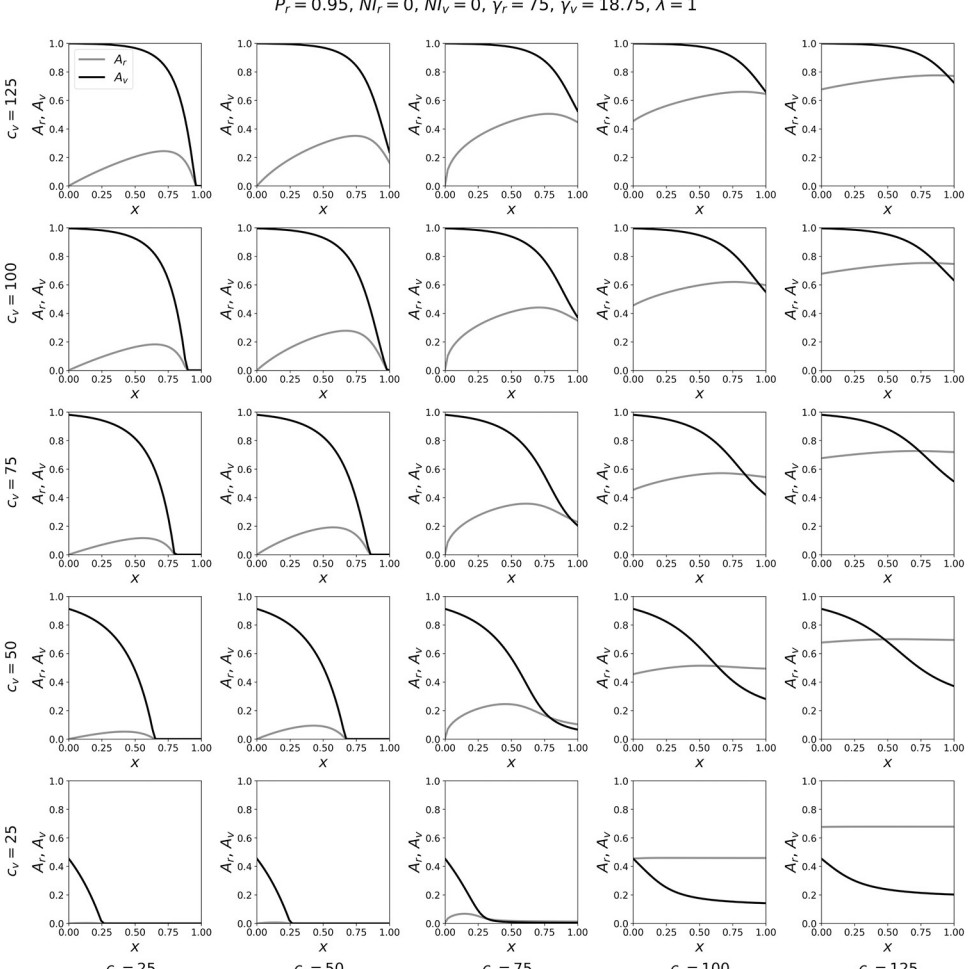

**Fig 3. Variation of attack-rates with x, for $\gamma_v = \gamma_r/4$.** Attack-rates $A_r$ and $A_v$ as functions of $x$, for a range of contact frequencies $c_r$ and $c_v$, for $\gamma_v = \gamma_r/4$.

For example, when $\gamma_v = 18.75$ (left column of panels), $c_r = 20$ and $c_v = 40$, the attack rate $A_v$ is large when $x = 0$. However, as $x$ is increased, the red contour lines shift upward, indicating a lowering of the attack rate at $(c_r, c_v) = (20, 40)$, until $A_v = 0$ (no epidemic in the $v$ population) in the second-last and last panels in the column ($x = 0.75$ and $x = 1$).

The positioning of the blue contour lines ($A_r$) is generally less affected by changes in $x$ than that of the red contours. This is particularly evident for the case of $\gamma_v = \gamma_r$ (right column of panels). This is due to the asymmetry in the sizes of the populations of the $r$ and $v$ groups ($N_v$ being 5% of the total population).

To better appreciate the model results summarized in the contour maps of Fig 2, it is helpful to simultaneously examine the attack rates for a particular point in the $(c_r, c_v)$ parameter-space as $x$ is varied. This is shown in Figs 3–5.

Figs 3–5 show the variation in the attack rates $A_r$ and $A_v$ as functions of $x$, for various $(c_r, c_v)$ coordinates. Each panel is for one pair of the $(c_r, c_v)$ coordinates, with $c_r$ increasing (in columns) from left to right, and $c_v$ decreasing (in rows) from top to bottom. In this way, one can visualize the behaviours of the attack rates with $x$, on the $(c_r, c_v)$ plane, across a range of $c_r$ and $c_v$ values sampled from the phase diagrams shown in Fig 2.

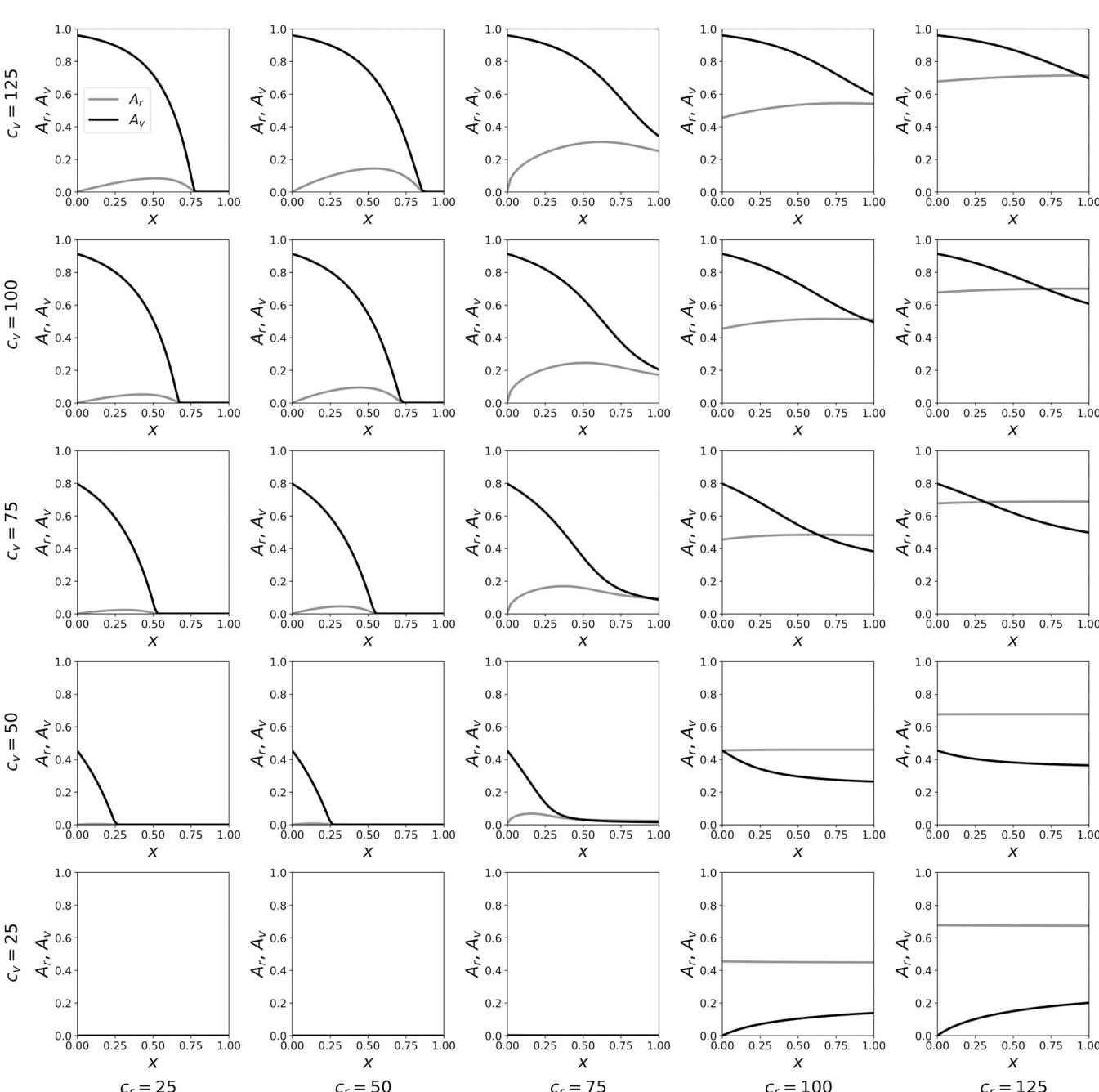

**Fig 4. Variation of attack rates with x, for $\gamma_v = \gamma_r/2$.** Attack-rates $A_r$ and $A_v$ as functions of $x$, for a range of contact frequencies $c_r$ and $c_v$, for $\gamma_v = \gamma_r/2$.

As can be seen, when $\gamma_v = \gamma_r/4$ (Fig 3), increasing $x$ decreases $A_v$ for all values of $(c_r, c_v)$ shown in the figure. The decrease in $A_v$ can be dramatic, including going from $A_v = 1$ for small values of $x$ to $A_v = 0$ for large values of $x$. Increasing $x$ generally increases $A_r$, and the increase in $A_r$ is largest for values of $c_r \leq \gamma_r$ (such that no epidemic would occur in the $r$ group if it were completely isolated from the $v$ group) and for intermediate values of $x$. In many of the panels of Fig 3, and especially when $c_r \leq \gamma_r$ and $c_v >> \gamma_v$, there is a maximum value of $A_r$ at a value of

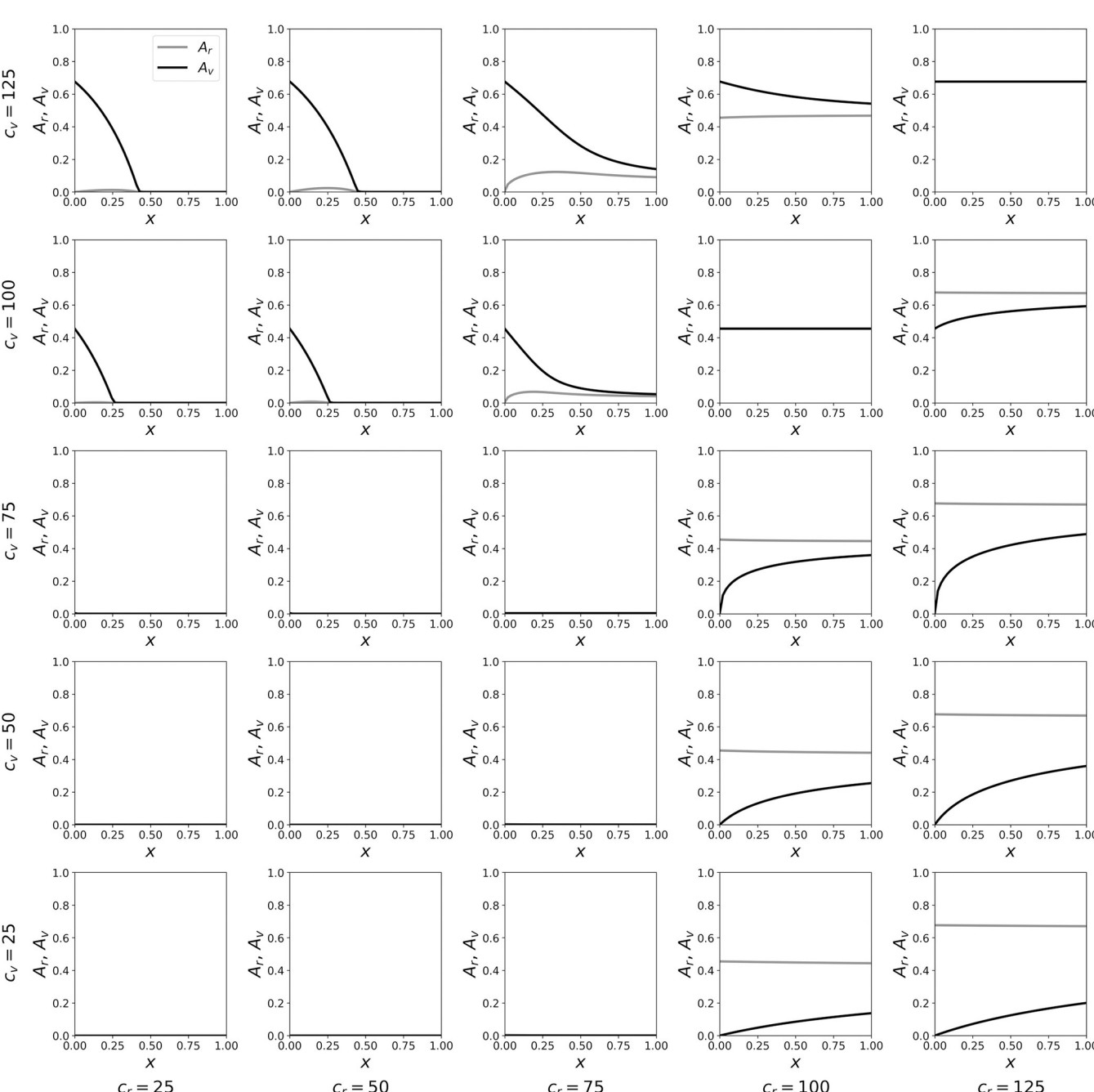

**Fig 5. Variation of attack rates with x, for $\gamma_v = \gamma_r$.** Attack-rates $A_r$ and $A_v$ as functions of $x$, for a range of contact frequencies $c_r$ and $c_v$, for $\gamma_v = \gamma_r$.

$x < 1$, which can be explained as follows. For $c_r \leq \gamma_r$ and $c_v >> \gamma_v$, and for $x$ close to 0, there is a large attack rate among the vulnerable population. Increasing $x$ above $x = 0$ causes some $r$ individuals to come into contact with $v$ individuals, who have a high probability of being infectious, thereby increasing $A_r$. However, as $x$ is increased further, the vulnerable become more diluted among the larger robust population, such that their attack rate decreases. The sharply decreasing attack rate among the vulnerable with increasing $x$ eventually produces conditions

in which the probability, during the course of the epidemic, that a susceptible robust person contacts an infectious person decreases, resulting in a decrease in $A_r$ as $x$ approaches 1. When $c_r > \gamma_r$, increasing $x$ has a very small effect on $A_r$, because the $r$ group has a much larger population than the $v$ group; this is also reflected in the small changes in the blue contour lines in Fig 2 for $c_r > \gamma_r$ and for increasing $x$.

When $\gamma_v = \gamma_r/2$ (Fig 4), increasing $x$ generally decreases $A_v$, similar to the results in Fig 3, and $x$ has a smaller effect on $A_r$ compared to the results in Fig 3. The only parameter values for which $A_v$ increases with $x$ are in the extreme lower-right corner of Fig 4, for which $(c_r, c_v) = (100, 25)$ and $(125, 25)$. For these two pairs of $(c_r, c_v)$ values, $c_v < \gamma_v$, such that the contact frequency of $v$ individuals is so low that an epidemic would not occur among the vulnerable if they were completely excluded from the majority group. Furthermore, for $(c_r, c_v) = (100, 25)$ and $(125, 25)$, $c_r$ is much greater than $c_v$, which is unrealistic given our interpretation of $c_{ij}$ implied by our simplifying assumption $\beta_{rr} = \beta_{vv} = \beta_{rv} = \beta_{vr} = 1$ (see the Model section). We note that a similar, small increase in $A_v$ versus $x$ also occurs in the case of $\gamma_v = 18.75$ when $c_v < \gamma_v$ and $c_r >> c_v$, as can be seen in the left column of panels in Fig 2, e.g. when $c_v \approx 15$ and $c_r = 120$.

When $\gamma_v = \gamma_r$ (Fig 5), $x$ has little effect on $A_r$, due to the differences in population sizes of the $r$ and $v$ groups. Increasing $x$ can decrease $A_v$ significantly when $c_v >> c_r$ (panels in the upper-left corner of Fig 5) and can increase $A_v$ significantly when $c_r >> c_v$ (panels in the lower-right corner of Fig 5). This asymmetry occurs because of the asymmetry in population sizes $N_r$ and $N_v$, causing $c_{rv} << c_{vr}$ (when $\lambda = 1$) such that it is much less likely for any given $r$ person to come into contact with a $v$ person than vice-versa. Similarly, in the right column of panels in Fig 2, increasing $x$ has a large effect on the red ($A_v$) contour lines and essentially no effect on the blue ($A_r$) contour lines.

In summary, increasing $x$ for fixed $c_v$ and $c_r$ decreases the attack rate in the vulnerable group across all realistic values of the contact frequencies, when $v$ represents a minority vulnerable population (here making up 5% of the total population and having a recovery time twice or four times as long as for the robust majority). This means that the vulnerable population is harmed by isolation from the robust population and benefits from mixing with or dilution within the robust population, in terms of risk of infection during the course of the epidemic or pandemic.

In S1 Appendix, we show that the same results hold when varying $P_r$, $\lambda$, and the seeding magnitude and distribution.

## Discussion

Using a general two-population epidemic model, we have shown that increasing the degree of intermingling of the minority vulnerable ($v$) population with the majority robust ($r$) population reduces the attack-rate among the vulnerable. The advantage to the vulnerable group of intermingling with the robust group increases as the vulnerability of the minority group increases, that is, as their disease recovery time increases. Increasing the share of a vulnerable person's interactions that are with other vulnerable people, by confining them together in the same facility, increases the likelihood of infection of the vulnerable person during the course of the epidemic or pandemic, because infected vulnerable people remain infectious for a long time, relative to robust people.

The only exception to this general rule occurs if the contact frequency for vulnerable individuals is so small that no epidemic would occur in the vulnerable group if it were completely segregated from the robust majority of society, while the frequency of guaranteed infection-causing contacts for robust people is large enough to produce an epidemic in that group and is

also much higher than that of vulnerable individuals. We expect this exception to be irrelevant in reality because it is unrealistic for $c_r >> c_v$, given the definition of the contact frequencies $c_{ij}$ as representing contacts of sufficient physical proximity and duration such that a susceptible $i$ person is guaranteed to be infected by an infectious $j$ person (see the Model section).

Our results also show that a trade-off can occur in which increasing $x$ (reducing segregation of $v$ from $r$, increasing intermingling of $v$ with $r$) causes $A_v$ to decrease and concomitantly causes $A_r$ to increase. This trade-off occurs in the region of parameter space where $\gamma_v << \gamma_r$, $c_v > \gamma_v$, and $c_r \le \gamma_r$ (e.g., see the panel of Fig 3 with $(c_r, c_v) = (75, 125)$). In this region of parameter space, no epidemic occurs in the $r$ population when it is completely separated from the $v$ population ($x = 0$). It is also the region of parameter space where increasing $x$ produces the largest decrease in $A_v$. Thus exposing the robust population to the vulnerable and thereby risking causing an epidemic among the robust, which would otherwise not occur in this limited region of the possible parameter space, is a desirable ethical choice (regarding the policy-imposed value of $x$) in a compassionate society, since vulnerable individuals are exponentially more at risk of dying than robust people from respiratory disease [49–51]. In other words, segregation ($x \rightarrow 0$) is contrary to protecting the vulnerable, including in circumstances in which there can be the said trade-off to the disadvantage of the robust who experience an epidemic.

Our analysis focuses on the two dominant and most fundamental features present in all epidemic models: the contact frequencies and recovery rates. On this simplest-possible yet sufficiently realistic foundation, we establish that segregating the vulnerable into care homes virtually always produces negative results in epidemic models. Not surprisingly, therefore, researchers using complex agent-based models have found that segregation of vulnerable individuals produces worse outcomes both for that group and for the society overall [52].

Others have used simple epidemiological models to study segregation of "high-transmission-risk" and "low-transmission-risk" groups [39, 53, 54]. However, because such studies are focused on different transmission rates due to different behavioural and contact characteristics of the two groups–such as sexual preferences, cultural lifestyle factors, and willingness to become vaccinated–they do not consider the impact of different recovery rates for the two populations, which is crucial in the context of segregation of vulnerable individuals from the robust majority. Those studies, therefore, do not directly address the problem of society's vulnerable sector regarding infectious diseases.

Segregation based on vaccination status has also been studied recently using simple models [55–58]. In this application, Hickey and Rancourt found that the effect of the segregation on increasing or decreasing the contact frequencies in the segregated groups is crucial and can cause the predicted epidemic outcomes to be worse for both the vaccinated and unvaccinated, compared to no segregation [55]. This highlights the importance of contact frequencies, which are necessarily impacted by segregation policies, and which again play a pivotal role in the present analysis.

Isolation policies intending to protect the vulnerable reduce their contacts with the outside world, for example by barring visitors from entering care homes and by reducing the frequency of interaction between care home staff and residents. The care home isolation policies are also designed to reduce the number of epidemiological contacts between the care home residents themselves. However, since transmission of respiratory diseases is air-borne via long-lived suspended aerosol particles [59, 60] and occurs in indoor environments [61], confining many vulnerable people in the same facility in-effect increases the per-individual frequency of infectious contacts, because they are breathing the same air and ventilation is imperfect. Indeed, virtually all studied outbreaks of viral respiratory illnesses have occurred in indoor environments [61–65] and care homes for the elderly are known to be "ideal environments" for outbreaks of infectious respiratory diseases, due to the susceptibility of the residents

living in close quarters [8, 11, 12]. A policy that decrease $c_{vr}$, for example by barring younger family members from entering care homes to visit their elderly relatives, causes the isolated vulnerable people to spend more time in the care home, breathing the same air as the other residents. This in-effect increases $c_{vv}$.

For constant $c_r$, decreasing $c_v$ reduces the attack rate in the vulnerable group, regardless of the value of $x$, as can be seen from Fig 2. However, the sought decreasing of $c_v$ is imposed by isolating the vulnerable (from society, loved ones and each other), which has important negative health consequences [6, 7, 66–69]. Psychosocial factors, including depression, lack of social support, and loneliness are known to play key roles in the negative health effects of isolation [70–74]. Proposed psychosocial factors uncovered by participatory qualitative research include dissonance between expectations and reality [75, 76], which could be significant for vulnerable elderly patients with no prior life experience relevant to the isolation measures applied during the COVID era, which had no historical precedent.

Whereas governments used theoretical epidemic models to justify most public health policies during the COVID era, within a tunnel vision of reducing risk of infection with a particular virus, they appear not to have considered what those same models predict about infection rates under conditions of care home segregation; and they appear to have disregarded the exponential increase of infection fatality rate with age [49–51]. Care home segregation policies may have been responsible for many deaths attributed to COVID-19 in Western countries.

We conclude that segregation and isolation of the vulnerable into care homes as a strategy to reduce the risk of infection during the course of an epidemic or pandemic is contrary to the most relevant immediate considerations from epidemiological models, in realistic conditions in which vulnerable people are highly susceptible and take longer to recover. The model parameter space, within possible parameter values, is one where it is virtually never epidemiologically advantageous to segregate and isolate frail people.

## Supporting information

**S1 Appendix. Additional results for different parameter values and seeding conditions.** (PDF)

## Author Contributions

**Conceptualization:** Joseph Hickey, Denis G. Rancourt.

**Data curation:** Joseph Hickey.

**Formal analysis:** Joseph Hickey, Denis G. Rancourt.

**Funding acquisition:** Joseph Hickey, Denis G. Rancourt.

**Investigation:** Joseph Hickey, Denis G. Rancourt.

**Methodology:** Joseph Hickey, Denis G. Rancourt.

**Project administration:** Joseph Hickey, Denis G. Rancourt.

**Resources:** Joseph Hickey, Denis G. Rancourt.

**Software:** Joseph Hickey.

**Validation:** Joseph Hickey, Denis G. Rancourt.

**Visualization:** Joseph Hickey, Denis G. Rancourt.

**Writing – original draft:** Joseph Hickey.

**Writing – review & editing:** Joseph Hickey, Denis G. Rancourt.

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
