## [Decision Letter · Decision Letter 0]

18 Aug 2023

PONE-D-23-16691Predictions from standard epidemiological models of consequences of segregating and isolating vulnerable people into care facilitiesPLOS ONE

Dear Dr. Hickey,

Thank you for submitting your manuscript to PLOS ONE. After careful consideration, we feel that it has merit but does not fully meet PLOS ONE’s publication criteria as it currently stands. Therefore, we invite you to submit a revised version of the manuscript that addresses the points raised during the review process.

We look forward to receiving your revised manuscript.

Kind regards,

Binod Acharya

Academic Editor

PLOS ONE

Journal Requirements:

Reviewers' comments:

Reviewer's Responses to Questions

**Comments to the Author**

1. Is the manuscript technically sound, and do the data support the conclusions?

Reviewer #1: Yes

Reviewer #2: Yes

2. Has the statistical analysis been performed appropriately and rigorously? 

Reviewer #1: N/A

Reviewer #2: N/A

3. Have the authors made all data underlying the findings in their manuscript fully available?

Reviewer #1: Yes

Reviewer #2: Yes

4. Is the manuscript presented in an intelligible fashion and written in standard English?

Reviewer #1: Yes

Reviewer #2: Yes

5. Review Comments to the Author

Reviewer #1: This article analyzes a two-population SIR model, studying the effect of isolating vulnerable populations. The authors find the counter-intuitive conclusion that isolation almost invariably leads to worse outcomes, as was seen in the Covid-19 pandemic. The study examines a relevant model with a thorough investigation of the key parameters: the intergroup contacts rates c and infection times gamma, as well as the infectiousness ratio lambda and initial distribution of infections (in the supplement).

There are a few minor changes that I think would improve this manuscript, roughly in order of importance.

1. The literature review should be expanded, there is a good chance similar models have been studied earlier.

2. If the authors found some intuition behind these findings, it would help to include this. For example, the issue seems to be that in this model the most vulnerable are also the most infectious, due to the longer recovery times, which explains why concentrating them is harmful.

3. Figures 3-5 are similar enough that it could be more effective to show just one and discuss how the plots change for different recovery times (perhaps with another figure showing the effect of varying gamma with other parameters constant).

4. In the inset of figure 1, it appears the attack rate of the robust population is non-monotonic, peaking around x = 0.8. This is surprising, as the other attack rate is strictly decreasing. It could be helpful to discuss this briefly.

5. It could help to relabel populations a and b with the letters r and v, as these would make it easier to remember which population is robust vs vulnerable.

Reviewer #2: Please see the attached file. .

6. PLOS authors have the option to publish the peer review history of their article (what does this mean?). If published, this will include your full peer review and any attached files.

Reviewer #1: No

Reviewer #2: No

---

## [Author Response · Author response to Decision Letter 0]

10 Sep 2023

Please see our file "Response to Reviewers.docx", which we uploaded as a separate file.

---

## [Decision Letter · Decision Letter 1]

16 Oct 2023

Predictions from standard epidemiological models of consequences of segregating and isolating vulnerable people into care facilities

PONE-D-23-16691R1

Dear Dr. Hickey,

We’re pleased to inform you that your manuscript has been judged scientifically suitable for publication and will be formally accepted for publication once it meets all outstanding technical requirements.

Kind regards,

Binod Acharya

Academic Editor

PLOS ONE

Additional Editor Comments (optional):

Reviewers' comments:

Reviewer's Responses to Questions

**Comments to the Author**

1. If the authors have adequately addressed your comments raised in a previous round of review and you feel that this manuscript is now acceptable for publication, you may indicate that here to bypass the “Comments to the Author” section, enter your conflict of interest statement in the “Confidential to Editor” section, and submit your "Accept" recommendation.

Reviewer #1: All comments have been addressed

Reviewer #2: All comments have been addressed

2. Is the manuscript technically sound, and do the data support the conclusions?

Reviewer #1: Yes

Reviewer #2: Yes

3. Has the statistical analysis been performed appropriately and rigorously? 

Reviewer #1: Yes

Reviewer #2: N/A

4. Have the authors made all data underlying the findings in their manuscript fully available?

Reviewer #1: Yes

Reviewer #2: (No Response)

5. Is the manuscript presented in an intelligible fashion and written in standard English?

Reviewer #1: Yes

Reviewer #2: Yes

6. Review Comments to the Author

Reviewer #1: (No Response)

Reviewer #2: (No Response)

7. PLOS authors have the option to publish the peer review history of their article (what does this mean?). If published, this will include your full peer review and any attached files.

Reviewer #1: No

Reviewer #2: No

---

## [Editor Report · Acceptance letter]

20 Oct 2023

PONE-D-23-16691R1 

Predictions from standard epidemiological models of consequences of segregating and isolating vulnerable people into care facilities 

Dear Dr. Hickey:

I'm pleased to inform you that your manuscript has been deemed suitable for publication in PLOS ONE. Congratulations! Your manuscript is now with our production department. 

Kind regards, 

on behalf of

Mr. Binod Acharya 

Academic Editor

PLOS ONE